# In Vivo Tau Burden Is Associated with Abnormal Brain Functional Connectivity in Alzheimer’s Disease: A ^18^F-Florzolotau Study

**DOI:** 10.3390/brainsci12101355

**Published:** 2022-10-06

**Authors:** Zizhao Ju, Zhuoyuan Li, Jiaying Lu, Fangyang Jiao, Huamei Lin, Weiqi Bao, Ming Li, Ping Wu, Yihui Guan, Qianhua Zhao, Huiwei Zhang, Jiehui Jiang, Chuantao Zuo

**Affiliations:** 1PET Center and National Research Center for Aging and Medicine & National Center for Neurological Disorders, Huashan Hospital, Fudan University, Shanghai 200035, China; 2School of Communication and Information Engineering, Shanghai University, Shanghai 200444, China; 3Department of Neurology, Huashan Hospital, Fudan University, Shanghai 200040, China; 4Institute of Biomedical Engineering, School of Life Science, Shanghai University, Shanghai 200444, China

**Keywords:** Florzolotau PET, functional connectivity, mild cognitive impairment, Alzheimer’s disease

## Abstract

Purpose: ^18^F-Florzolotau is a novel second-generation tau radiotracer that shows higher binding affinity and selectivity and no off-target binding. The proportion loss of functional connectivity strength (PLFCS) is a new indicator for representing brain functional connectivity (FC) alteration. This study aims to estimate the relationship between the regional tau accumulation and brain FC abnormality in Alzheimer’s disease (AD) and mild cognitive impairment (MCI) patients based on Florzolotau PET and fMRI. Methods: 22 NC (normal control), 31 MCI and 42 AD patients who have already been scanned with ^18^F-Florzolotau PET were recruited in this study. (We calculated the PLFCS and standardized uptake value ratio (SUVR) of each node based on the Brainnetome atlas (BNA) template. The SUVR of 246 brain regions was calculated with the cerebellum as the reference region. Further functional connection strength (FCs), PLFCS and SUVR of each brain region were obtained in three groups for comparison.) For each patient, PLFCS and standardized uptake value ratio (SUVR) were calculated based on the Brainnetome atlas (BNA) template. These results, as well as functional connection strength (FCs), were then compared between different groups. Multiple permutation tests were used to determine the target nodes between NC and cognitive impairment (CI) groups (MCI and AD). The relationship between PLFCS and neuropsychological scores or cortical tau deposit was investigated via Pearson correlation analysis. Results: Higher PLFCS and FCs in AD and MCI groups were found compared to the NC group. The PLFCS of 129 brain regions were found to be different between NC and CI groups, and 8 of them were correlated with tau SUVR, including superior parietal lobule (MCI: *r* = 0.4360, *p* = 0.0260, AD: *r* = −0.3663, *p* = 0.0280), middle frontal gyrus (AD: MFG_R_7_2: *r* = 0.4106, *p* = 0.0129; MFG_R_7_5: *r* = 0.4239, *p* = 0.0100), inferior frontal gyrus (AD: IFG_R_6_2: *r* = 0.3589, *p* = 0.0316), precentral gyrus (AD: PrG_R_6_6: *r* = 0.3493, *p* = 0.0368), insular gyrus (AD: INS_R_6_3: *r* = 0.3496, *p* = 0.0366) and lateral occipital cortex (AD: LOcC _L_4_3: *r* = −0.3433, *p* = 0.0404). Noteworthily, the opposing relationship was found in the superior parietal lobule in the MCI and AD groups. Conclusions: Brain functional connectivity abnormality is correlated with tau pathology in AD and MCI.

## 1. Introduction

Alzheimer’s disease (AD) is an irreversible, devastating neurodegenerative disorder characterized by the aberrant accumulation and aggregation of amyloid plaques and neurofibrillary tangles in the brain [1]. Mild Cognitive Impairment (MCI) is identified as a prodromal phase of AD, with 10–15% conversion to dementia per year [2]. While as the most common dementia, available treatments of AD can only relieve clinical symptoms, none of the interventions are able to slow down its development. Therefore, the pathological mechanism and early diagnosis of AD have attracted increasing interest. Tau deposition and spreading are major factors of AD, which exist early in the cascade of AD etiopathogenesis and result in neuronal loss and cognitive decline [3]. Thus, it is considered an ideal target for diagnosis and novel treatments. To evaluate the pathological tau burden in vivo, a wide variety of tracers were developed based on the diverse binding targets of tau-paired helical filaments(PHFs), including quinoline derivatives and benzimidazole pyrimidine derivatives [4,5,6,7,8]. Since the first tau radioligand, ^18^F-FDDNP, was developed, a variety of tracers have been synthesized and have demonstrated promising results in the clinical evaluation of tau deposition. However, the first-generation tau tracers showed several limitations, including high binding affinity in the deep brain nucleus, where pathological studies did not show a high density of tangles in AD and “off-target” binding to monoamine oxidase B (MAO-B). Compared with them, the second-generation tau tracers showed lower “off-target” binding and improved affinity and selectivity in tau aggregates [9], which enhanced the detection of the affected subregions in the early phase of AD. Several clinical trials have verified that the distribution of tau tracers is related to post-mortem neuropathology in primary tauopathies [10], and the binding of tracers is associated with cognitive performance in AD patients [10]. Moreover, tau PET imaging can contribute to further elaboration on the relationship between tau accumulation and other biomarkers or clinical symptoms.

Brain functional connectivity (FC) is able to evaluate the spatiotemporal association between distinct cerebral cortical regions, which can offer novel perspectives on functional brain disruption and other abnormalities caused by neuropathies [11]. In several neurodegenerative diseases, neuropathology and atrophy are most prominent in nodes with dense connections (usually referred to as ‘hubs’) [12], both at the structural [13] and functional levels [14]. Abnormal functional connectivity strength (FCs) in AD and its preclinical stages have been estimated via resting-state functional magnetic resonance imaging (rs-fMRI) in previous studies [15,16]. Several studies were launched to evaluate the correlation between functional abnormality and tau deposition in AD and its preclinical stages [17,18,19,20]. Combined with tau positron emission tomography (PET) scan, studies provided evidence that tau toxicity can influence neuronal activity and synaptic plasticity and lead to the disruption of FC. For instance, Hansson et al. proposed that there were spatial correspondences between major functional networks and regional pathological tau accumulation [21,22]. Cope et al. has demonstrated that tau burden is correlated with a higher graph theoretical index of functional connectivity evaluated in AD [22]. Specifically, the proportion loss of functional connectivity strength (PLFCS) was assessed, which ensured that the results did not issue from the bias introduced by proportionate thresholding. This index is associated with weighted degree, while it is more subject to fMRI signal-to-noise ratio limitations in comparison to traditional FC metrics. Previous studies showed that the PLFCS is a relatively new index of research, and it holds possibilities in revealing the brain function change in AD and other neurodegenerative diseases [23,24,25]. However, all these studies were launched with the use of first-generation tau tracers.

In this study, we used the second-generation tau radiotracer ^18^F-Florzolotau to evaluate the relationship between the abnormal FC (including functional connection strength and PLFCS) and tau deposition, as well as neuropsychological scores in AD and MCI, and further explored the role of tau accumulation in the FC abnormality during AD disease progression.

## 2. Materials and Methods

### 2.1. Participants

We prospectively recruited 36 AD patients, 26 MCI patients and 22 NC subjects who underwent Florzolotau PET scanning in Huashan Hospital affiliated with Fudan University in this study. Besides Florzolotau PET, T1-weighted structural MRI and fMRI scanning was also needed for each subject. All participants needed to be over 55 years and fulfill the following research criteria: (1) AD patients have completed an ^18^F-AV45 PET scan and been characterized as amyloid PET-positive; (2) subjects have conducted The Mini-Mental State Examination (MMSE) and Clinical Dementia Rating–Sum of Boxes (CDR-SB) test by experienced neurologists from Huashan hospital; (3) NCs had an MMSE score of 24 or greater, and no history of cognitive impairment, mental disorders, neurological diseases or brain trauma. Clinically probable AD was diagnosed with the 2011 NIA-AA guidelines [24], and the diagnosis of MCI required to meet Petersen’s criteria [26]; (4) dementia caused by other reasons needs to be excluded. 

This study was approved by the Institutional Review Board of Huashan Hospital (HIRB) (no. 2018-363). All subjects or a legally responsible relative gave written informed consent before the study.

### 2.2. Acquisition Protocol

All subjects withdrew cognitive enhancing and psychotropic medicine for at least 12 h before clinical assessment and each imaging acquisition. Participants underwent T1-weighted structural MRI and rs-fMRI on a 3.0T horizontal magnet (Discovery MR750; GE Medical Systems, Milwaukee, WI) and a T1 MRI image was acquired with FOV = 25.6 cm, matrix = 256 × 256 × 152, slice thickness = 1 mm, repetition time (TR) = 8.2 ms, echo time (TE) = 3.2 ms, flip angle= 12°. The Rs-fMRI scans were performed with the following parameters: FOV = 24 cm, slice thickness = 3 mm, TR = 8800 ms, TE = 145 ms, flip angle = 77°. PET data were obtained by the use of a Siemens mCT Flow PET/CT (Siemens, Erlangen, Germany) in a PET center, Huashan Hospital, in the three-dimensional (3D) mode. ^18^F-Florzolotau PET imaging was performed for 20 minutes, 90–110 minutes after 370 MBq ^18^F-Florzolotau was intravenously injected. Images were corrected by the mode of CT attenuation correction, and the reconstruction was performed with the ordered subset expectation maximization (OSEM) method.

### 2.3. Data Pre-Processing

Rs-fMRI data were processed with the use of DPARSF (http://www.rfmri.org/DPARSF, accessed on 10 January 2022). First, in order to stabilize the initial signal and allow individuals to acclimate to the environment, the first 10 volumes were discarded. Volumes remaining according to acquisition time were corrected and realigned to the head movement of the first volume, and mean signals from white matter (WM), cerebrospinal fluid (CSF) and Friston-24 head motion parameters were regressed out. Then, the T1 images were registered with the fMRI at the individual level. The segmented T1 images were spatially normalized based on the standard Montreal Neurological Institute (MNI) brain space. All images were resampled into 3 × 3 × 3 mm^3^ voxels. Finally, the linear drift and corrections for white matter, CSF signals, six head movement parameters and band-pass filters (0.01–0.08 Hz) of the fMRI data were removed. Smoothing was based on a 4 mm full-width half-height (EWHM) filter.

PET data were processed using Statistical Parametric Mapping 12 (the Wellcome Department of Neurology, London, UK) package. First, PET images were registered based on the T1 images of the corresponding subjects. Second, the gray matter (GM) tissue probability map from segmented T1 images was registered. Then, based on the MNI standard space, the GM map was registered using nonlinear transformation parameters. The registered PET images were also spatially normalized using the same transformation parameters and were then resampled into 2 × 2 × 2 mm^3^ voxels. Finally, PET was smoothed based on an 8 mm full-width half-height filter.

### 2.4. Brain Network Analysis 

#### 2.4.1. Functional Connection Strength

The Brainnetome atlas (BNA) contains 246 brain regions of the bilateral hemispheres (available at http://atlas.brainnetome.org/, last accessed on 31 August 2022). Each brain region was treated as a node used for network analysis based on Pearson’s correlation to estimate time-resolved fMRI connectivity. Thus, each subject obtained a 246 × 246 association matrix and Fisher z-transform. The individual connection strength of each node was quantified by the sum of the absolute values of the associated value between the node and other nodes, then the connection strength of the node was defined as the average value of the individual connection strength for each diagnostic group.

#### 2.4.2. Proportional Loss of Connectional Strength

The individual-level PLFCS in the disease group was defined as the difference in connectivity strength from the normal control group and was scaled based on the baseline group. The value of PLFCS was calculated as equation (1):(1)$${\mathrm{Loss}}_{i}=\frac{\mu_{i}-\sigma }{\sigma }$$
where $\mu_{i}$ is the connection strength of each node. σ is the average connection strength of all nodes in the baseline [23,27].

### 2.5. Semi-Quantitative ROI-Based PET Analyses

The entire cerebellum was selected as the reference region for calculating the normalized uptake value ratios (SUVRs) in 246 brain regions based on PET images. Then, for comparison with functional connection strength, the group-averaged SUVR of each brain region was obtained.

### 2.6. Comparison of PLFCS along AD Spectrum

Except for these three groups, we define the cognitive impairment (CI) group as the sum total of AD and MCI patients. To determine robust PLFCS biomarkers, we performed a two-sample *t*-test for PLFCS in the 246 brain regions of the NC and the CI group. We considered brain regions with significant differences (*p* < 0.05) as potential biomarkers and presented them using boxplots.

### 2.7. Correlation between Proportional Loss and Clinical Scales

We first checked the relationship between the PLFCS and functional connection strength (un-transformed) of 246 brain regions in the disease groups to evaluate whether brain regions with high connection strength were vulnerable to the effect of neuropathology, based on our hypotheses that the proportionate vulnerability of these regions can reflect the progression of neurodegeneration. It was expected that regions with relatively higher connection strength tend to lose more. Therefore, we performed a correlation analysis between the averaged proportional loss of hub regions and cognitive level in the disease groups, where the cognitive level was reflected by clinical scales of MMSE and CDR-SB.

### 2.8. Statistical Analysis

A two-sample *t*-test was performed for continuous variables comparison, and the χ test was used for the between-group differences of categorical variables. Correlations between fMRI data and tau levels and clinical scales were assessed for each diagnostic group based on Pearson correlation. We analyzed differences of clinical variates among three groups with the use of univariate analysis of variance (ANOVA) and Bonferroni’s post hoc analysis or Dunn’s multiple test on account of homogeneity of variance. All statistical analyses were performed on the SPSS 19.0 platform, and *p* < 0.05 was considered significant.

## 3. Results

### 3.1. Demographic

The subject’s demographics and scores of cognitive examination are displayed in Table 1. The average age of the MCI group was older than NC and AD, whereas the difference between NC and AD subjects was not statistically significant (*p* = 0.226). Lower MMSE and higher CDR-SB scores were observed in AD and MCI than NC group. No differences in education years were found among the three groups.

### 3.2. Results of PLFCS and Tau Level

Higher FCs and PLFCS were found in the MCI and AD groups than in the NC group, and differences were not found between the AD and MCI groups (FCs: *p* = 0.971, Figure 1a; PLFCS: *p* = 0.652, Figure 1b). Higher globally averaged tau SUVR was found in the AD group than in the MCI and NC groups, and differences were not found between the NC and MCI groups (*p* = 0.887, Figure 1c).

We found differences in the PLFCS level between the NC and CI groups in 129 brain regions and further analyzed the correlation between PLFCS and tau SUVR in these regions. Brain regions with significant correlation are shown in Table 2 and Figure 2, including superior parietal lobule (MCI: *r* = 0.4360, *p* = 0.0260, AD: *r* = −0.3663, *p* = 0.0280), middle frontal gyrus (AD: MFG_R_7_2: *r* = 0.4106, *p* = 0.0129; MFG_R_7_5: *r* = 0.4239, *p* = 0.0100), inferior frontal gyrus (AD: IFG_R_6_2: *r* = 0.3589, *p* = 0.0316), precentral gyrus (AD: PrG_R_6_6: *r* = 0.3493, *p* = 0.0368), insular gyrus (AD: INS_R_6_3: *r* = 0.3496, *p* = 0.0366) and lateral occipital cortex (AD: LOcC_L_4_3: *r* = −0.3433, *p*= 0.0404). Noteworthily, the inverse relationship was found in the superior parietal lobule in the MCI and AD groups. Further, we analyzed the relationship between SUVR/PLFCS and clinical scores, and no significant correlation was found.

### 3.3. Correlation of FC and PLFCS

The correlation between the group-averaged FCs and PLFCS of the cerebral cortex of the MCI and AD groups with that of the NC group are shown in Figure 2. We found that brain regions with higher functional connectivity in normal control subjects lost the larger proportion of connection strength in MCI (*r* = −0.7736, *p* < 0.0001, Figure 3a) and AD (*r* = −0.7999, *p* < 0.0001, Figure 3b).

## 4. Discussion

The study aimed to investigate the neurophysiological FC abnormality in AD and MCI and its relationship with tau burden. We found higher PLFCS and functional connection strength in the AD and MCI groups compared to NCs. Global tau SUVR of the AD group was higher than that of the NC group, but no difference was found between the AD and MCI groups. Significant relationships between the FCs of NC groups and the PLFCS of disease groups were observed, which strongly supported the hypotheses that brain regions with relatively higher functional connectivity would be more vulnerable to tau accumulation in neurodegeneration [28]. PLFCS of several brain regions were found to be different between the NC and CI groups, and some of them were correlated with tau SUVR, including superior parietal lobule, middle frontal gyrus, inferior frontal gyrus, precentral gyrus, insular gyrus and lateral occipital cortex. Notably, relationships were found in the superior parietal lobule in both MCI and AD groups. 

Consistent with our results, Cho et al. found that ^18^F-flortaucipir SUVR increased in the superior parietal cortex, and the change was correlated with the progression of diffuse volume atrophy during a 2-year-follow-up in the MCI group, and the correlation pattern with clinical scores was not included [29]. The cortical thickness of the superior parietal lobule was found to be associated with mild symptoms and signs of cognitive impairment in AD [30], indicating that the functional and structural abnormality in the superior parietal lobule were correlated with disease progression but did not directly influence the cognitive performance of patients. Strikingly, in the AD group, we found that PLFCS was negatively correlated with tau SUVR, and in MCI, this relationship was inversed, with the value of PLFCS increasing in line with the tau level. This may be a compensatory phenomenon. The cognitive function of healthy aging depends on maintaining connectivity within and between large-scale networks [31], which can increase the fault tolerance of the network to disease [32]. By the time AD pathology is sufficiently advanced to trigger the clinical signs of MCI, tau has generally already emerged to some degree throughout the neocortex [33], thus the functional connections of some nodes are strengthened to balance the weakening of those strongest functional connections, which are caused by tau pathology [34].

We found a negative relationship between FCs/PLFCS and the tau level in the occipital and parietal lobes and a positive relationship in the frontal lobe; however, both tau and FCs/PLFCS were not associated with cognitive function (evaluated by MMSE and CDR-SB) in these brain regions. Several studies have reported the same fronto-occipital functional alteration pattern and demonstrated that this change was associated with cognition and tau level [33,34,35,36], suggesting a reconstructed brain network, including further disconnection and isolation in parietal and occipital nodes and compensatory frontal network with the progression of AD. Our PLFCS findings not only consolidated those results but also indicated that the local tau accumulation might be a reason for the network’s alteration in AD progression, and there was no interactive effect of the tau burden with functional connectivity alteration on cognitive function. To our knowledge, this is the first time these have been verified with a second tau PET tracer, and the results are consistent with previous studies, which show convincing evidence of the value of second-generation tau tracers Florzolotau in studies on AD continuum.

Though the average level of PLFCS in AD subjects was prominently higher than that of NCs, the PLFCS value cannot distinguish AD from MCI. Considering that both aging and neurodegeneration contribute to the disruption of the functional network, the higher PLFCS level in the MCI group may result from their older age [37]. A previous study found that it was age-related that the connectivity in the MCI group was significantly stronger than that in the NC group [38]. During the aging and memory deficit process, an evident reduction in the connection strength was demonstrated as attributed to factors such as the massive loss of neurons and synapses during the development of aging and AD [9]. 

Besides the regions shown in the current study, several studies have also reported that regional connectivity alteration might be a potential biomarker of the AD continuum and relate to proteinopathies. Li et al. found that FCs decreased in the left MTG of MCI patients, particularly in the vascular MCI (VaMCI) [39]. Our previous study performed individually specified PLFCS to quantitatively characterize the decrease degree and the PLFCS of the left middle temporal gyrus that yielded a powerful diagnostic efficacy of 80% for categorizing SCD from NC. Moreover, significant gray matter volume reductions, altered FC pattern and density, together with severely decreased amplitude of low-frequency fluctuation in the left MTG have been reported in previous studies on MCI [40,41,42,43]. These findings indicated that the MTG was a crucial network node with rich connections, and it could be a promising target in studies of the AD continuum. In addition to MTG, other regional connectome alterations such as hippocampal connectivity [17] and frontoparietal control network hubs [43] were also reported on MCI and AD, and some were proven to be related to tau pathology in disease progression [44]. Therefore, other FC assessments would be used to estimate the functional abnormality and its relationship with tau burden, and additional regional connectome influenced by cognition impairment-related pathological alterations would be analyzed in further studies, which might offer new insights into the pathological interpretation of the relationship between tau accumulation and the functional connectivity and find more potential biomarkers in the AD continuum.

This study had several shortcomings. First, the sample size was relatively small and statistical differences in age and MMSE scores were found between groups, which limited our interpretation of causality. Secondly, the AD group included patients with different severity, which might influence the uptake levels of Florzolotau in brain regions and further result in different relationships with PLFCS. Meanwhile, only the MMSE and CDR-SB scores were used to estimate the cognitive function of AD and MCI patients. More subdomain evaluations, such as memory, executive function and others, are needed so that we can evaluate the cognitive changes in AD patients more accurately.

## 5. Conclusions

In summary, we have shown the negative relationship between FCs/PLFCS and tau level in the occipital and parietal lobes and a positive relationship in the frontal lobe, suggesting that the local tau accumulation might be a reason for the functional alteration and network reconstruction during the AD progression. An inverse relationship between PLFCS and tau SUVR in SPL was found in the MCI and AD groups, indicating that a compensatory functional strengthening may exist in some nodes in response to the regional tau toxicity. These findings provide preliminary evidence that brain FC alteration is associated with tau pathology in AD and its prodromal stage and motivates the exploration of AD physiopathology with the combination of rs-fMRI and PET imaging with a second-generation tau tracer.

## Figures and Tables

**Figure 1 brainsci-12-01355-f001:**
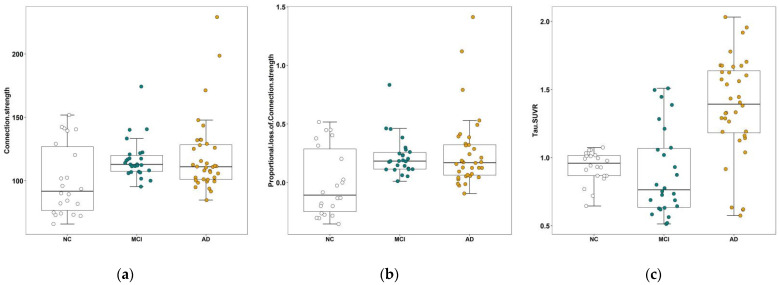
Comparison of connection strength, PLFCS and tau SUVR between the three groups. (**a**) The global FCs of NC, MCI and AD groups. (**b**) The comparison of global PLFCS between NC, MCI and AD groups. (**c**) The comparison of global tau SUVR between the NC, MCI and AD groups. Abbreviations: NC, normal control; MCI, mild cognitive impairment; AD, Alzheimer’s disease.

**Figure 2 brainsci-12-01355-f002:**
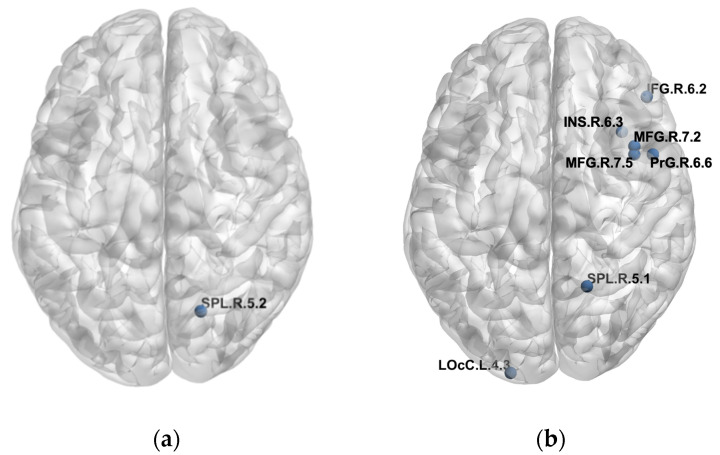
Detailed brain regions with significant correlation of FCs/PLFCS and tau SUVR. (**a**) Brain regions with significant correlation of FCs/PLFCS and tau SUVR in the MCI group. (**b**) Brain regions with significant correlation of FCs/PLFCS and tau SUVR in the AD group.

**Figure 3 brainsci-12-01355-f003:**
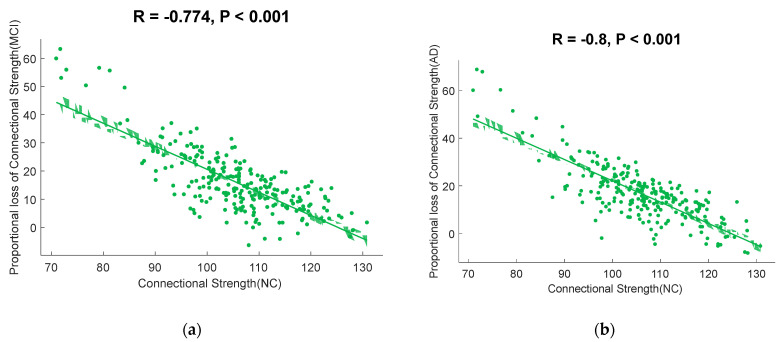
Scatterplots illustrating the relationship between the connectional strength in normal controls and PLFCS in disease groups. (**a**) The correlation between the connectional strength in normal controls and the PLFCS in MCI subjects. (**b**) The correlation between the connectional strength in normal controls and PLFCS in AD subjects.

**Table 1 brainsci-12-01355-t001:** Demographic information and scores of cognitive examinations.

Group	Number	Gender (Male/Female)	Age of Scanning	Education	MMSE	CDR-SB
NC	22	8/14	56.95 ± 7.01	11.27 ± 3.92	28.23 ± 1.41	0.00 ± 0.00
MCI	26	6/20	70.38 ± 8.47	10.27 ± 3.29	25.53 ± 1.70	4.46 ± 1.46
AD	36	14/22	60.94 ± 10.13	9.86 ± 3.86	16.00 ± 6.29	8.68 ± 3.96
*p*	-	0.217 ^a^	<0.001 ^b^	0.358 ^b^	<0.001 ^c^	<0.001 ^b^

^a^ Chi-square test. ^b^ One-way ANOVA test with Bonferroni’s multiple comparison test. ^c^ ANOVA test with Dunn’s multiple comparison test. *p* < 0.05 was considered as significant. *p*-values are given for the comparisons among the three groups. Data are presented as mean ± standard deviation. AD, Alzheimer’s disease; CDR-SB, Clinical Dementia Rating–Sum of Boxes; NC, normal control subjects; MCI, patients with mild cognitive impairment; MMSE, Mini-Mental State Examination.

**Table 2 brainsci-12-01355-t002:** Brain regions with significant correlation of FCs/PLFCS and tau SUVR.

MCI											
Lobe	Gyrus	Left and Right Hemispheres	Label ID.L	Label ID.R	Modified Cyto-Architectonic	lh.MNI (*X*, *Y*, *Z*)	rh.MNI (*X*, *Y*, *Z*)	R(FCs and Tau SUVR)	P (FCs and tau SUVR)	R (PLFCS and Tau SUVR)	P (PLFCS and Tau SUVR)
Parietal lobe	SPL, superior parietal lobule	SPL_R_5_2		128	A7c, caudal area 7	−15, −71, 52	19, −69, 54	0.4332	0.0271	0.4360	0.0260
**AD**											
**Lobe**	**Gyrus**	**Left and Right Hemispheres**	**Label ID.L**	**Label ID.R**	**Modified Cyto-Architectonic**	**lh.MNI (** ** *X* ** **,** ** *Y* ** **,** ** *Z* ** **)**	**rh.MNI (** ** *X* ** **,** ** *Y* ** **,** ** *Z* ** **)**	**R (FCs and Tau SUVR)**	**P (FCs and Tau SUVR)**	**R (PLFCS and Tau SUVR)**	**P (PLFCS and Tau SUVR)**
Frontal lobe	MFG, middle frontal gyrus	MFG_R_7_2		18	IFJ, inferior frontal junction	−42, 13, 36	42, 11, 39	0.4159	0.0116	0.4106	0.0129
	MFG, middle frontal gyrus	MFG_R_7_5		24	A8vl, ventrolateral area 8	−33, 23, 45	42, 27, 39	0.4214	0.0105	0.4239	0.0100
	IFG, inferior frontal gyrus	IFG_R_6_2		32	IFS, inferior frontal sulcus	−47, 32, 14	48, 35, 13	0.3656	0.0283	0.3589	0.0316
	PrG, precentral gyrus	PrG_R_6_6		64	A6cvl, caudal ventrolateral area 6	−49, 5, 30	51, 7, 30	0.3420	0.0412	0.3493	0.0368
Parietal lobe	SPL, superior parietal lobule	SPL_R_5_1		126	A7r, rostral area 7	−16, −60, 63	19, −57, 65	−0.3624	0.0299	−0.3663	0.0280
Insular lobe	INS, insular gyrus	INS_R_6_3		168	dIa, dorsal agranular insula	−34, 18, 1	36, 18, 1	0.3321	0.0478	0.3496	0.0366
Occipital lobe	LOcC, lateral occipital cortex	LOcC_L_4_3	203		OPC, occipital polar cortex	−18, −99, 2	22, −97, 4	−0.3443	0.0397	−0.3432	0.0404

## Data Availability

Not applicable.

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
