# Peer review of "In Vivo Tau Burden Is Associated with Abnormal Brain Functional Connectivity in Alzheimer’s Disease: A 18F-Florzolotau Study"

_brainsci, 2022, doi:10.3390/brainsci12101355_

Round 1
Reviewer 1 Report
In the manuscript entitled “In vivo Tau burden is associated with abnormal brain functional connectivity in Alzheimer’s disease: a [18F]-Florzolotau study”, Zizhao Ju and colleagues showed that the study evaluated the relevance between functional connectivity and tau retention in Alzheimer's disease (AD) patients by combining the proportion loss of functional connectivity strength (PLFCS) based on resting-state functional magnetic resonance imaging (fMRI) and [18F]-Florzolotau PET imaging. Based on the result, it came to a conclusion that Tau pathology is associated with brain functional connectivity abnormality in AD and mild cognitive impairment (MCI). This study used a reasonable and reliable method and proved the brain connection relative to tau retention. However, I cannot recommend publication in the current form due to the following additional issues which should be addressed.
1. In the abstract, the statement of the conclusion “Tau pathology is associated with brain functional connectivity abnormality in AD and MCI” was inaccurate. The hypothesis of this study was the tau retention may lead to abnormal function of the brain and the result proved the relationship of tau retention and brain functional connectivity abnormality. So the conclusion needed to be modification.
2. It had not showed clearly in the method whether this study was a retrospective study or a prospective study.
3. In the method, the drugs of cognitive enhancer or psychotropic medicine in MCI subjects was not clearly state. If MCI took those medicines, it should influence the brain metabolism which might lead to an inaccurate conclusion.
I believe that after minor revisions, this research paper may have good quality.
Author Response
Thank you for giving us the opportunity to revise and re-submit our manuscript. We are also grateful to the editor and reviewers for their insightful comments, suggestions, and questions, which, we believe, have helped to significantly strengthen our article. The responses to comments from the reviewer were as follow.
1. Thank you for your comment, which we agree with. We have modified the statement as "Brain functional connectivity abnormality is correlated with tau pathology in AD and MCI" in line 38-39 and "These findings provide preliminary evidence that brain FC alteration is associated with tau pathology in AD and its prodromal stage" in line 335-336.
2. Thank you for pointing this out. The data was retrospectively analyzed but the study was designed prospectively. And we have amended in the revised manuscript in line 99 as "We prospectively recruited 36 AD patients, 26 MCI patients and 22 NC subjects who underwent Florzolotau PET scanning in Huashan Hospital affiliated to Fudan university in this study. "
3. Thank you for highlighting this. All the MCI subjects enrolled in this study were not treated with cognitive enhancer or psychotropic medicine. We added a statement to the revised manuscript in line115-116.
Reviewer 2 Report
Reading the manuscript entitled: In vivo Tau burden is associated with abnormal brain functional connectivity in Alzheimer’s disease: a [18F]-Florzolotau study was really interesting. The author aims to investigate the neurophysiological FC abnormality in AD and MCI and its relationship with Tau burden.
The manuscript is scientifically sound and the experimental design is suitable for testing the hypothesis proposed by the authors. The material and method section describes in an easy-to-follow way the experimental protocol applied by the authors.
There are a lot of typos issues. Other typos include lack of spaces before brackets (e.g.: can proveide, decline[3], tauopathies[11], patients[12], stages[19-22], Hansson et al.proposed…, tau accumulation[23,24], FC(including functional…, etc
- Please add the appropriate reference: Cope et al. has demonstrated…;
- The entire article should be reviewed by a native English speaker. There are quite a few phrases that are difficult to understand (e.g.:51-55, 56-65).
Author Response
We regret our mistakes in punctuation errors and incoherent sentences. We revised our manuscript and modified the descriptions. For example, in line 50-65 it was changed in the revised manuscript as "Therefore, the pathological mechanism and early diagnosis of AD have attracted increasing interests. Tau deposition and spreading are major factors of AD, which exist early in the cascade of AD etiopathogenesis and result in neuronal loss and cognitive decline [3]. Thus, it is considered ideal target for diagnosis and novel treatments. To evaluate pathological tau burden in vivo, a wide variety of tracers were developed based on the diverse binding targets of tau paired helical filaments(PHFs), including quinoline derivatives and benzimidazole pyrimidine derivatives [4-9]. Since the first tau radioligands 18F-FDDNP was developed, a variety of tracers were synthesized and have been demonstrated promising in the clinical evaluation of tau deposition. However, the first-generation tau tracers showed several limitations, including high binding affinity in the deep brain nucleus where pathological studies did not show a high density of tangles in AD and “off-target” binding to monoamine oxidase B (MAO-B). Compared with them, the second-generation tau tracers showed lower “off-target” binding and improved affinity and selectivity in tau aggregates [10], which enhanced the detection of the affected subregions in the early phase of AD. Several clinical trials have verified that the distribution of tau tracers is related to post-mortem neuropathology in primary tauopathies."
Reviewer 3 Report
The paper is well written it contains 2 tables and 2 figures, 1 equation, and all the paper sections there are no major spelling/grammar errors
the authors shown the negative relationship between FCs/PLFCS and tau 319 level in occipital and parietal lobe, and positive relationship in frontal lobe, suggesting 320 that the local tau accumulation might be a reason of the functional alteration and network 321 reconstruction during the AD progression. thus it can be accepted for publication as is
Author Response
Thank you very much for your positively responses to our study and its value.